# Combined Analysis of Second- and Third-Generation Transcriptome Sequencing for Gene Characteristics and Identification of Key Splicing Variants in Wound Healing of Ganxi Goat Skin

**DOI:** 10.3390/ani14213085

**Published:** 2024-10-26

**Authors:** Xue Yang, Lucheng Zheng, Junhong Huo, Wei Hu, Ben Liu, Qingcan Fan, Wenya Zheng, Qianqian Wang

**Affiliations:** 1College of Life Science and Resources and Environment, Yichun University, Yichun 336000, China; zhenglc@jxycu.edu.cn (L.Z.); ycxyhuwei@163.com (W.H.); liubenres@163.com (B.L.); huoshan8fan@126.com (Q.F.); zhengwenyares@163.com (W.Z.); xiaoxiaoqian_qian@126.com (Q.W.); 2Institute of Animal Husbandry and Veterinary, Jiangxi Academy of Agricultural Science, Nanchang 330200, China; hjh_0222@126.com

**Keywords:** Ganxi goat, transcriptome sequencing, skin, wound healing, splicing variants

## Abstract

Ganxi goat (*Capra hircus*) is a unique breed of goat in Jiangxi Province, China, and plays an important role in local animal husbandry. Skin plays an indispensable role as the main barrier protecting the body from external damage. However, little is known about the skin and molecular regulation mechanism of wound healing in Ganxi goats. In this study, the full-length transcriptome data of Ganxi goat skin and splicing variant events of their core wound healing genes provide a positive significance for enriching the information annotation of the goat genome and studying the mechanism of skin wound healing.

## 1. Introduction

Wound healing and skin regeneration are critical biological processes, particularly in the context of livestock breeding and veterinary medicine. Efficient wound healing can significantly impact the health, productivity, and welfare of livestock, such as goats, which are valuable for their meat, milk, and fiber. Understanding these processes at a molecular level is essential for developing targeted interventions that can enhance animal health and breeding outcomes. Moreover, insights gained from studying wound healing in animals can be translated into medical applications for humans, contributing to advancements in regenerative medicine and therapeutic interventions for skin-related conditions.

Full-length transcriptomics was selected for this study because it provides a comprehensive view of gene expression regulation, especially in the context of complex biological processes like wound healing. Traditional transcriptome sequencing techniques like RNA-Seq can only measure partial transcripts, whereas full-length transcriptomics can capture complete transcript information, including 5′ and 3′ UTRs and splicing variants [1]. Therefore, full-length transcriptomics is crucial for studying the gene expression regulation and RNA splicing in complex genomes.

This advanced method has been widely applied in species characterization and disease research, highlighting its potential to uncover novel regulatory mechanisms. For instance, Bai Y et al. used full-length transcriptome sequencing to study the dorsal skin of the Chinese giant salamander, generating a high-quality full-length reference gene set to elucidate its genetic characteristics and identify functional skin proteins [2]. Xue X et al. treated severe plaque psoriasis patients with oxymatrine and conducted full-length transcriptome sequencing on their lesion skin samples, discovering that oxymatrine might alter the abnormal expression of certain genes and pathways in psoriasis, with *CDK1* and *CCNB1* being identified as potential therapeutic targets [3]. Zhang L et al. analyzed the expression characteristics of long non-coding RNAs (LncRNAs) and messenger RNAs (mRNAs) in human traumatic brain injury using full-length transcriptome sequencing, finding differential expression profiles post-injury [4]. Zhang H et al. generated full-length transcriptome data from rumen (*n* = 2) and testis (*n* = 1) samples using PacBio Iso-Seq technology, improving the gene annotation of the goat genome [5].

Splice variants are an important regulatory mechanism in gene expression, producing various protein isoforms from a single gene through different combinations of exons in precursor mRNA molecules. This alters protein structures and diversifies their functions [6]. The study of splice variants is significant for understanding gene expression regulation, disease mechanisms, drug development, and biological evolution, drawing widespread attention and research [7,8]. In livestock, understanding these mechanisms is crucial not only for improving animal health, but also for selecting traits that enhance wound healing and skin regeneration, which can lead to better breeding outcomes. This manuscript builds on our previous work by further investigating whether the splice variants of hub genes play a role in the wound healing process [9]. These hub genes—*CDC20*, *MMP2*, *TIMP1*, and *EDN1*—were identified in our earlier studies using a wound healing model specifically developed for Ganxi goats. We established this model and conducted transcriptome sequencing at various time points after injury. Through enrichment analysis and screening, we identified these genes as crucial contributors to wound healing. These genes play crucial roles in cell cycle regulation, cell migration, matrix degradation, tissue repair, and disease progression, which are essential for understanding biological processes and disease development.

This study employs transcriptomics to examine the mRNA expression profiles in Ganxi goat skin, utilizing combined second- and third-generation sequencing technologies. By focusing on wound healing, a process vital for maintaining skin integrity and overall animal health, this research aims to enrich goat genome annotations and elucidate splice variant events of relevant hub genes. The findings could inform selective breeding strategies that would enhance wound healing in livestock and provide a foundation for developing novel therapies in veterinary and human medicine.

## 2. Materials and Methods

### 2.1. Samples and Data Collection

Three 1-year-old female Ganxi goats, weighing 18.1 ± 0.68 kg, were purchased from Yuanzhou District, Yichun City, Jiangxi Province, and housed under the same environmental conditions. Following anesthesia and disinfection, a 5 cm full-thickness skin incision was made on the upper middle right flank. Immediately, a 2 cm^2^ skin tissue sample was collected (day 0 post-injury). Subsequently, adjacent skin tissue samples were collected at the wound edge on day 5 post-injury using the same sampling and preservation methods. The samples were preserved in RNA Keeper Tissue Stabilizer (Vazyme, Nanjing, China) at 4 °C for 24 h and then transferred to −20 °C for storage. All experimental procedures were approved by the Animal Research Ethics Committee of Yichun University.

### 2.2. Second-Generation Transcriptome Sequencing (Illumina Sequencing)

The QC method for Illumina sequencing and the related data can be referred to in our previous study [9].

### 2.3. Construction and Analysis Workflow of Third-Generation Full-Length Transcriptome Libraries

The total RNA quality was assessed by checking the concentration and purity using the NanoDrop (Thermo Scientific, Waltham, MA, USA), and integrity was evaluated using the Agilent 2100 Bioanalyzer (Agilent, Palo Alto, CA, USA) with the RNA 6000 Nano kit (Agilent, Palo Alto, CA, USA). Samples were required to have a concentration of >300 ng/μL and an RNA Integrity Number (RIN) of >8.0. RNA meeting these criteria was reverse transcribed and amplified using the SMARTer PCR cDNA Synthesis (Clontech, Palo Alto, CA, USA) and PrimeSTAR GXL DNA Polymerase kit (Clontech, Palo Alto, CA, USA). The resulting cDNA was purified using magnetic beads and divided into the following two aliquots: one purified with 1* magnetic beads and the other with 0.4* magnetic beads. Equimolar mixing of the purified cDNA aliquots was then performed for library construction. Library construction involved using the Template Prep Kit 1.0 (Pacific Biosciences, Menlo Park, CA, USA). DNA damage repair and end repair were conducted initially, followed by purification. Adapters were ligated to form a dumbbell-shaped library. Enzymatic digestion was employed to remove failed adapter-ligated fragments, followed by library purification using magnetic beads. Before sequencing, library binding was carried out using the Sequel Binding and Internal Ctrl Kit 3.0 and MagBead Binding Buffer Kit v2 (Pacific Biosciences), followed by primer and polymerase binding. The library concentration for sequencing was 8–9 pmol. The workflow of this experiment is illustrated in Figure 1.

### 2.4. Sequencing

Both ordinary transcriptome and full-length transcriptome sequencing were performed by Nanjing Personal Gene Technology Co., Ltd (Shanghai, China). The raw data were uploaded to the SRA database (https://www.ncbi.nlm.nih.gov/sra/PRJNA922517, (accessed on 20 June 2024)) under the accession number PRJNA922517. Eukaryotic reference transcriptome analysis was conducted using the following genome reference from the Ensembl database: Capra_hircus.ARS1.dna.toplevel.fa (https://asia.ensembl.org/Capra_hircus, (accessed on 5 September 2024)).

### 2.5. Third-Generation Sequencing Data Processing

Each raw sequence was divided into one or multiple subreads, meaning there were multiple subreads per zero-mode waveguide (ZMW). All subreads within the same ZMW originated from the same transcript. Due to the random nature of base calling errors, the PacBio official isoseq3 analysis pipeline was used to process the sequencing data, and subreads from the same ZMW were corrected to obtain Read of Insert (ROI, Circular Consensus Sequencing reads) sequences. These ROI sequences were differentiated based on sequencing adapters at both ends and polyA sequences at the ends to obtain full-length non-chimeric sequences. These sequences underwent clustering and polishing to obtain high-quality isoform (transcript) sequences. Statistics were performed separately for each sample’s ROI, including the number of ROI sequences, total bases in the ROIs, average length of ROI sequences, average quality of ROI sequences, and classification statistics of the ROIs, including full-length non-chimeric sequences, full-length chimeric sequences, and non-full-length sequences, among other categories. Full-length non-chimeric sequences were clustered and corrected using the isoseq3 cluster and polish software (and based on second-generation data, the corrected data were subjected to a second round of correction using proovread based on second-generation sequencing data) to obtain HQ isoforms. Subsequent analyses were based on these HQ isoforms.

### 2.6. Genome Annotation Analysis

#### 2.6.1. Alignment of HQ Isoforms to Reference Genome

The HQ isoform sequences were utilized as complete transcript sequences for analysis. The minimap2 tool was employed to align these HQ isoform sequences to the reference genome using the following parameters: -ax splice -t 30 -uf --secondary=no -C5. The alignment results revealed the matching status between the HQ isoforms and the genome. Despite the HQ isoforms analyzed by isoseq3 being relatively accurate, some redundant sequences may still have existed. Therefore, the collapse_isoforms_by_sam.py tool from the cDNA_Cupcake (17.0.0, free) software package was used to further cluster and merge the aligned sequences. This process helped to eliminate similar transcripts, thereby reducing redundancy, and provided alignment information to better understand these merged transcripts.

#### 2.6.2. Isoform Classification and Annotation

SQANTI3 (3.0, free) software was utilized to analyze the deduplicated transcripts and compare them with the reference genome annotation to investigate the relationship between the sequenced isoforms and known genes and transcripts. Additionally, isoforms were filtered based on second-generation sequencing data to further remove low-confidence transcripts. SQANTI3 classified isoforms into eight types based on their alignment positions and splicing information compared to known transcripts (Figure 2), as follows:

FSM (Full Splice Match): Isoforms that completely matched the splice sites of known transcripts.

ISM (Incomplete Splice Match): Isoforms that partially matched the splice sites of known transcripts but exhibited splice differences.

NIC (Novel In Catalog): Isoforms with splice sites inconsistent with known transcripts but present in the known genome annotation.

NNC (Novel Not in Catalog): Isoforms with splice sites inconsistent with known transcripts and absent from the known genome annotation.

Antisense: Isoforms opposite in direction to known transcripts.

Genic Intron: Isoforms located within introns of genes.

Genic Genomic: Isoforms that overlapped with known genes but did not match known transcripts.

Intergenic: Isoforms unrelated to known genome annotation, located in intergenic regions.

#### 2.6.3. Isoform Open Reading Frame Prediction

The Transdecoder (v5.5.0, free) software was employed to predict open reading frames (ORFs) for the merged isoform sequences.

#### 2.6.4. Functional Annotation Analysis of Isoforms

Multiple databases were utilized to annotate the functions of the isoforms, including NR (NCBI non-redundant protein sequences), GO (Gene Ontology), KEGG (Kyoto Encyclopedia of Genes and Genome), eggNOG (evolutionary genealogy of genes: Non-supervised Orthologous Groups), and Swiss-Prot. The specific annotation steps were as follows. For NR Annotation, the isoform sequences were compared to protein sequences in the NR database using the diamond tool to obtain the similarity and functional information between the isoform sequences of this species and protein sequences of closely related species. For GO Annotation, the eggNOGmapper (2.1.6, free) software was utilized, mapping the GO annotation results to GOTerm and counting the number of annotations to isoforms at the second-level classification. For KEGG Annotation, KOBAS was employed to complete KO (KEGG Orthology) and Pathway annotation. After KO annotation, KO was mapped to the corresponding KEGG Pathway to aid in understanding the metabolic pathways and functions in which isoforms were involved. For eggNOG Annotation, the eggNOGmapper software was used for the eggNOG annotation of isoforms. Each isoform was classified into the eggNOG classification directory based on its correspondence with the eggNOG classification catalog, further categorizing and understanding its function. Through these functional annotation steps, a comprehensive understanding of the functions of isoforms and their roles in the cell can be obtained.

#### 2.6.5. LncRNA Analysis

The identification of lncRNAs was carried out using a combination of multiple tools and approaches to ensure the accurate classification and prediction of novel lncRNAs. Initially, isoforms were annotated using the SQANTI3 software, where isoforms classified as lncRNAs were identified as known lncRNA isoforms based on existing gene annotations. For isoforms annotated as novel genes, we employed a multi-step approach to predict their coding potential. First, we utilized the CPC2 (Coding Potential Calculator v2) software to assess the coding potential of each novel isoform. CPC2 evaluates both the sequence and structural features to reliably distinguish between coding and noncoding RNA. Isoforms predicted to lack coding potential were classified as candidate novel lncRNAs. To further validate the noncoding nature of these novel isoforms, we cross-referenced the predictions with additional tools such as CNCI (Coding-Non-Coding Index) and Pfam-scan, which assess coding potential based on different parameters. This cross-validation helped to reduce false positives and improve the robustness of novel lncRNA identification. Furthermore, we filtered the candidate lncRNAs based on transcript length (>200 nucleotides) and the absence of significant open reading frames (ORFs) to ensure consistency with lncRNA characteristics.

#### 2.6.6. Analysis of Alternative Splicing Events

A single gene can exhibit multiple alternative splicing forms, significantly increasing the gene’s coding capacity and information content. However, abnormal alternative splicing forms may have harmful or beneficial effects on organisms under specific environmental conditions, providing more pathways for the evolution of eukaryotic genes. To explore these alternative splicing events, the SpliceGrapher software (0.2.7, free) was used to predict and plot the alternative splicing models of isoforms. Additionally, alternative splicing events of isoforms were statistically analyzed. This analysis helped to reveal the regulatory mechanisms and biological significance of gene alternative splicing under different conditions.

#### 2.6.7. Fusion Gene Analysis

A fusion gene refers to the process where the entire or partial sequences of two genes merge to form a new gene. The formation of fusion genes may result from chromosomal translocations, partial deletions, or chromosomal inversions, among other chromosomal structural changes. Typically, fusion genes possess oncogenic potential. Gene fusion is a common phenomenon in tumors, promoting tumor initiation and progression, and sometimes serving as a molecular diagnostic and therapeutic target for cancer. In the comparison results with the genome, fusion_finder.py was used to extract sequences with alignments to multiple genes and with a certain level of sequencing support. These sequences were considered as potential fusion gene candidates. Subsequently, these fusion gene candidates were annotated to further analyze their characteristics and potential roles.

#### 2.6.8. Analysis of Alternative Polyadenylation (APA)

The APA of mRNA is a crucial post-transcriptional event in eukaryotic cells, widely present across various eukaryotic organisms. Since the 3′ end of mRNA typically contains cis-elements crucial for mRNA stability, localization, and translation, APA can be considered as an important transcriptional regulatory mechanism. APA was conducted using TAPIS, with the following filtering criteria set: reads support greater than 0 and the requirement for a minimum distance of 15 bp between adjacent polyadenylation sites.

#### 2.6.9. Validation of RNA-Seq Data Using Quantitative Real-Time PCR (qRT-PCR)

The total RNA extracted from the skin of Ganxi goats, which was used for RNA-Seq sequencing, was subjected to qRT-PCR. Primers were designed using the PrimerPremier 6.0 software, and the primer sequences are listed in Table 1. The differential expression of candidate genes was validated by qRT-PCR, with *GAPDH* selected as the reference gene. The expression levels were calculated using the 2^−ΔΔCt^ method. For RNA-Seq data analysis, the calculation method was as follows: the number of reads mapped to each candidate gene in each sample divided by the number of reads of the reference gene *GAPDH*, with the expression levels relative to the control group (CK) being calculated. Each sample included three biological replicates, and each biological replicate comprised three technical replicates. After setting up the PCR reaction system (Table 2) and selecting the cDNA template from each sample for the qRT-PCR reaction, the qRT-PCR reaction was performed using the prepared reaction system. It was placed on the real-time PCR instrument and the following reaction program was run: initial denaturation at 95 °C for 5 min, denaturation at 95 °C for 15 s, annealing at 60 °C for 30 s, and extension (40 cycles). The relative expression level of PCR products was calculated using the 2^−ΔΔCt^ method. The formula for calculating relative expression level is the following: ΔCT = CT (target gene) − CT (reference gene), where CT represents the threshold cycle number.

### 2.7. Identification and Analysis of Splice Variants

#### 2.7.1. Pattern Identification of Splice Variants

A single gene can exhibit multiple alternative splicing forms, significantly enhancing the potential and diversity of gene coding. Abnormal alternative splicing forms may have harmful or beneficial effects on organisms under specific environmental conditions, providing more opportunities for gene evolution in eukaryotes. Utilizing the third-generation sequencing data, the SpliceGrapher software was employed to predict and illustrate the alternative splicing models of isoforms. A structural analysis of splice variants was conducted, including their total length, number of exons, CDS length, alignment to the reference genome, and splicing patterns. PBID refers to the sequence identification number obtained from the third-generation sequencing data, with each PBID representing a unique sequence. In this experiment, PBID was used as the nomenclature for splice variants.

#### 2.7.2. Analysis of Cumulative Number of Splice Variants

Combining the second-generation sequencing data, splice variants were quantified by calculating Junction counts, representing the reads spanning splice junctions. The exon inclusion level (ϕ) was utilized to quantify alternative splicing, where ϕ = (I/LI)/(I/LI + S/LS), indicating the percentage of transcripts containing the alternative splicing event region among those including and skipping the event region. The Δϕ (Δϕ = |ϕ1 − ϕ2|) and FDR values were employed to determine the differential alternative splicing between two groups of samples, where ϕ1 and ϕ2 represent the exon inclusion levels of the two sample groups. When Δϕ > 5% and FDR ≤ 1%, it was considered that differential alternative splicing occurred at that splice site between the two sample groups. Combining third-generation sequencing data, reads spanning splice junctions from the second-generation sequencing counts were aligned to the sequences of corresponding splice variants for each gene, and the number of each splice variant was counted for expression analysis.

### 2.8. Data Statistical Analysis

GraphPad Prism 8.0 application software was utilized for data analysis, with the results presented as mean ± standard error of the mean (SEM). Significance analysis was performed using SPSS 19.0, with * indicating significant differences (*p* < 0.05) and ** indicating extremely significant differences (*p* < 0.01).

## 3. Results

### 3.1. Total RNA Quality Assessment

The total RNA extracted from the skin samples of Ganxi goats was pooled in equal amounts, and the mixed sample was subjected to agarose gel electrophoresis. The experimental results are shown in Figure 3A, where, from top to bottom, they represent 28S, 18S, and 5S rRNA, respectively. The ratio of 28S to 18S was approximately 2:1, with a clear, single, and bright main band, indicating that the RNA quality met the requirements for subsequent experiments. The mixed total RNA sample was further subjected to quality assessment using the Agilent 2100 Bioanalyzer and the Agilent 5067-4626 High-Sensitivity DNA Kit. The quality assessment results are summarized in Table 3. The sample concentration was measured at 474.47 ng/μL, with an RNA Integrity Number (RIN) of nine, indicating an excellent RNA integrity. This met the library construction requirements for full-length transcriptome sequencing (PacBio), with a total RNA concentration of ≥300 ng/μL and RIN of ≥7.5, ensuring the requirements for subsequent library construction and sequencing.

### 3.2. Sequencing Results and Data Analysis

#### 3.2.1. PacBio Sequencing Data Statistics

The statistical results indicate that a total of 1,010,225 ROI sequences were generated in this sequencing run, with a cumulative base count of 2,367,860,756. The lengths of the ROI sequences were predominantly distributed between 1000 and 3000 bp, with an average length of approximately 2343 bp (refer to Table 4 and Figure 3B). Moreover, the quality values of the ROI sequences were mostly above 0.95, with an average quality value close to 1 (refer to Figure 3C), indicating a high level of accuracy in the ROI sequences, thus ensuring the reliability of the sequencing data.

#### 3.2.2. ROI Sequence Classification Statistics

In this experiment, a total of 1,010,225 ROI sequences were detected, among which 82,968 were non-full-length sequences, 590 were full-length chimeric sequences, and 926,667 were full-length non-chimeric sequences. Within the full-length non-chimeric sequences, 1223 did not contain a polyA tail, while 925,444 did contain a polyA tail (Appendix A).

#### 3.2.3. HQ Isoform Statistics

This process yielded 56,981 high-quality sequences (HQ isoforms) with an accuracy greater than 99%. The total number of bases in high-quality sequences with an accuracy greater than 99% was 132,393,079. Subsequent analyses were based on HQ isoforms.

### 3.3. Genome Annotation

#### 3.3.1. Alignment Annotation of Isoforms to the Goat Genome

The alignment results showed that, out of 56,981 HQ isoforms, 52,791 aligned to the reference genome, with a total alignment rate of 92.65%. Among them, 1302 regions aligned less than 99%, accounting for 2.47% (Appendix A). Annotations were made to 10,907 annotated genes, with 2834 new genes being identified (Appendix A). A total of 3794 sites were optimized based on the annotation information of the original genome (Appendix A). There is still significant room for research on new genes, providing basic data for goat gene studies. The SQANTI3 classification statistics showed that isoforms were annotated to eight types of transcription forms, including 8412 FSMs, 2933 ISMs, 2936 NICs, 15,233 NNCs, 276 Antisense, 3 Genic Introns, 1497 Genic Genomics, and 2589 Intergenic (Appendix A). The transcript annotation distribution was also calculated for different lengths, with most transcripts ranging from 1000 to 3000 bp, with FSMs and NNCs being the primary forms (see Figure 3D for detailed results).

#### 3.3.2. Prediction of Open Reading Frames (ORFs) in Isoforms

A total of 30,947 isoforms were predicted to have ORFs, out of which 22,200 isoforms were predicted to have complete ORF structures. Additionally, 1680 isoforms were predicted to have ORFs, but lacked a stop codon at the 3′ end, while 6774 isoforms were predicted to have ORFs, but lacked a start codon at the 5′ end. Moreover, 293 isoforms were predicted to have only internal sequences (Appendix A).

#### 3.3.3. Results of Functional Annotation Analysis of Isoforms

The results revealed that 30,460 isoforms (98.43%) were annotated against the NR database, which represented the highest proportion. The eggNOG database annotated 30,190 isoforms (97.55%), while the KEGG database annotated the fewest isoforms at 22,265 (71.95%). Additionally, 29,188 isoforms (94.32%), 26,842 isoforms (86.74%), and 29,729 isoforms (96.06%) were annotated against the GO, Pfam, and Swiss-Prot databases, respectively. Moreover, 20,066 isoforms (64.84%) were annotated in all seven databases (Appendix A).

##### NR Annotation Statistics

In the NR database, a total of 30,460 Unigenes were annotated. Among them, 55.8% were annotated to *Capra hircus*, 15.76% to *Ovis aries*, 5.7% to *Bos taurus*, 2.16% to *Oryx dammah*, 1.57% to *Bubalus bubalis*, 1.38% to *Bos mutus*, and 0.8% to *Odocoileus virginianus texanus* (Appendix A).

##### GO Annotation Statistics

In the GO database, a total of 29,188 unigenes were annotated, primarily concentrated in biological processes such as cellular processes (26,583), the regulation of biological processes (21,536), and metabolic processes (18,673). They were also found in cellular components such as cell structures (28,625) and protein-containing complexes (11,428), as well as in molecular functions like binding (22,266) and catalytic activity (11,203). Detailed results are provided in Figure 4.

##### KEGG Annotation Statistics

In the KEGG database, a total of 22,265 isoforms were annotated, distributed across 35 secondary classification pathways in 5 main biological metabolism pathways. These pathways included metabolism (12), genetic information processing (5), environmental information processing (3), cellular processes (5), and organismal systems (10). The most abundant isoforms were found in the signal transduction pathway (3596) within environmental information processing, followed by the immune system pathway (2527) within organismal systems, and then the transport and catabolism pathway (2201) within cellular processes. Detailed results are provided in Figure 5.

##### eggNOG Annotation Statistics

The statistical results showed that 30,190 isoforms were annotated and classified into 24 categories. Among these, isoforms annotated as unknown function were the most abundant (7436), followed by those involved in signal transduction mechanisms (6111), transcription (2912), post-translational modification, protein turnover, and chaperones (2553). These results indicate that the genes in Ganxi goats exhibit significant species specificity, and their functions may have undergone some degree of divergence during evolution, providing valuable directions for exploring the functions of unknown genes. See Figure 6 for detailed results.

#### 3.3.4. Results of LncRNA Analysis

The results revealed the identification of 32 known LncRNAs (Appendix A) and 256 new LncRNAs (Appendix A), providing reference data for research on LncRNAs.

#### 3.3.5. Alternative Splicing Events Analysis

The results showed that there were a total of 12,283 alternative splicing events in the skin transcriptome of Ganxi goats, with exon skipping being the most frequent (3954 events), followed by intron retention (2861 events), and the least frequent being alternative 3′ end events (2495 events) (Appendix A).

#### 3.3.6. Results of Fusion Gene Analysis

A total of 112 fusion genes were detected, among which 45 had known genes on both ends, 11 had novel genes on both ends, 34 had a novel gene at the front, and 22 had a novel gene at the rear. The chromosomal distribution of these fusion genes is shown in Figure 7.

#### 3.3.7. APA Analysis

A total of 549 genes with APA were identified. Among them, there were 132 genes with more than 5 polyadenylation sites, 32 with 5 polyadenylation sites, 52 with 4 polyadenylation sites, 63 with 3 polyadenylation sites, 118 with 2 polyadenylation sites, and 152 with 1 polyadenylation site (Appendix A).

### 3.4. Identification and Statistical Analysis of CDC20 Splice Variants

#### 3.4.1. Identification of *CDC20* Splice Variants

Analysis of the third-generation sequencing results revealed the presence of three splice variant forms of *CDC20*. A visual model was established (as shown in Figure 8A), and its structural information was analyzed. Among them, two were splice variant forms with changes at the 5′ and 3′ ends, and one was a novel splice variant form with at least one new splice site. However, the CDS sequences of the three splice variants were consistent, with a length of 1500 bp (see Table 5).

#### 3.4.2. Validation of *CDC20* Transcriptome Data by qPCR and Analysis of Expression Levels of Its Splice Variants

The reliability of the *CDC20* transcriptome data was validated using qPCR, with goat GAPDH as the reference gene. Fluorescence quantitative PCR was performed using the same total RNA samples from Ganxi goat skin as used in the RNA-Seq sequencing. As shown in Figure 8B, the expression trend of *CDC20* in qPCR was consistent with the sequencing results, confirming the authenticity and reliability of the *CDC20* transcriptome data.

Analysis of the second- and third-generation sequencing results revealed the number of *CDC20* splice sites and analyzed the expression levels of each splice variant. The results showed no significant differences in the expression levels among the splice variant forms within the group. Compared to the skin samples at 0 days after injury (CK group), the expression levels of each *CDC20* splice variant increased in the skin samples at 5 days after injury (W group). The new splice variant form PB.1241.2 showed a significant increase, while the expression differences of the other two splice variant forms were not significant (Figure 8C).

### 3.5. Identification and Statistical Analysis of MMP2 Splice Variants

#### 3.5.1. Identification of *MMP2* Splice Variants

Analysis of the third-generation sequencing results revealed the presence of seven splice variants of *MMP2*. A visualization model was constructed (Figure 9A), consisting of five known splice forms and two newly discovered splice forms. Among the five known splice forms, four exhibited changes at both the 5′ and 3′ ends, while one displayed incomplete splicing at the 3′ end as a new splice variant. The two newly discovered splice forms both featured at least one new splicing site. Three distinct CDS sequences were obtained, with lengths of 1986 bp, 1755 bp, and 1140 bp, respectively (Table 6).

#### 3.5.2. Validation of *MMP2* Transcriptome Data by qPCR and Analysis of Expression

##### Levels of Its Splice Variants

The reliability of the *MMP2* transcriptome data was verified using qPCR, with goat *GAPDH* serving as the internal reference gene. Fluorescent quantitative PCR was conducted using the same total RNA samples from Ganxi goats’ skin as those used in RNA-Seq sequencing. The results, as depicted in Figure 9B, demonstrate that the expression trends of *MMP2* observed via qPCR were consistent with the sequencing results, confirming the authenticity and reliability of the *MMP2* transcriptome data.

Analysis of the second- and third-generation sequencing results involved quantifying the number of *MMP2* splice sites and analyzing the expression levels of each splice variant. The results indicate that there were no significant differences in the expression levels of the splice variants within each group. In comparison to the skin samples from 0 days post-injury (CK group), the expression levels of all *MMP2* splice variants increased in the skin samples from 5 days post-injury (W group). Notably, three splice variant forms, including PB.7960.2, PB.7960.3, and PB.7960.4, exhibited significant increases in their expression levels (Figure 9C).

### 3.6. Identification and Statistical Analysis of TIMP1 Splice Variants

#### 3.6.1. Identification of *TIMP1* Splice Variants

Analysis of the third-generation sequencing results revealed the presence of two splice variants of *TIMP1*. Visual models were constructed to illustrate these variants (Figure 10A). Both splice variants were novel forms, with one being a novel splice variant with at least one new splice site, and the other being a novel splice variant with intron retention. Two CDS sequences were obtained, with lengths of 624 bp and 438 bp, respectively (Table 7).

#### 3.6.2. Validation of *TIMP1* Transcriptome Data via qPCR and Analysis of Its Splice Variant Expression Levels

To validate the *TIMP1* transcriptome data, qPCR was employed using goat *GAPDH* as the internal reference gene. Fluorescence quantitative PCR was conducted using the total RNA samples from Ganxi goat skin, which were the same as those used for the RNA-Seq sequencing. The results, depicted in Figure 10B, demonstrate that the qPCR expression trend of *TIMP1* was consistent with the sequencing results, affirming the authenticity and reliability of the *TIMP1* transcriptome data.

Furthermore, analysis of the second- and third-generation sequencing results involved quantifying the number of *TIMP1* splice sites and analyzing the expression levels of each splice variant. The findings reveal that, compared to the skin samples taken at 0 days post-injury (CK group), the expression levels of all *TIMP1* splice variants significantly increased in the skin samples taken at 5 days post-injury (W group). Among these variants, the splice variant PB.12178.1 was predominant, and notably, its expression in the skin samples at 0 days post-injury was significantly higher than that of the splice variant PB.12178.2 within the same sample group (Figure 10C).

### 3.7. Identification and Statistical Analysis of Splice Variants of EDN1

#### 3.7.1. Identification of Splice Variants of *EDN1*

Analysis of the third-generation sequencing results revealed the presence of three splice variants of *EDN1*. A visualization model was constructed (Figure 11A), showing two known splice variant forms and one newly discovered splice form. Both known splice variant forms involved changes at the 5′ and 3′ ends, while the newly discovered splice form resulted in two distinct CDS sequences with lengths of 609 bp and 411 bp, respectively (Table 8).

#### 3.7.2. Validation of EDN1 Transcriptome Data by qPCR and Analysis of Splice Variant Expression Levels

To validate the transcriptome data of *EDN1*, qPCR was performed using the total RNA samples from goat skin, with *GAPDH* as the reference gene, consistent with the RNA-Seq sequencing. The results are presented in Figure 11B, showing that the qPCR expression trends of *EDN1* were generally consistent with the sequencing results, confirming the reliability of the *EDN1* transcriptome data.

Analysis of the second and third-generation sequencing results involved quantifying the number of splice sites of *EDN1* and assessing the expression levels of each splice variant. The analysis revealed that, compared to the skin samples at 0 days post-injury (CK group), the expression levels of all *EDN1* splice variants decreased in the skin samples taken at 5 days post-injury (W group), although these differences were not statistically significant (Figure 11C).

## 4. Discussion

Wound healing and skin regeneration are critical biological processes with significant implications in livestock breeding and veterinary medicine. Efficient wound healing not only enhances the health and productivity of livestock, but also has potential applications in improving breeding strategies and reducing economic losses. Furthermore, insights from wound healing in animals can inform medical research, contributing to the development of new treatments and therapies in human medicine, particularly in regenerative medicine and the management of chronic wounds. To further enrich the genetic information database of Ganxi goats, this study employed the PacBio sequencing platform to perform full-length transcriptome sequencing of Ganxi goat skin. A total of 40 Gb of base data were obtained, comprising 1,010,225 ROI sequences with an average length of 2343 bp. The quality values were all above 0.95, with an average quality value close to 1. These results indicate that the full-length transcriptome sequencing obtained transcripts with a good integrity and high quality, suitable for downstream analyses such as functional annotation and splice variant identification.

After further filtering and analysis, we obtained 926,667 non-chimeric full-length sequences. Through clustering and correction against the goat genome, we obtained 56,981 high-quality sequences with an accuracy greater than 99%. Pan et al. conducted the full-length transcriptome sequencing of muscle tissues from Lingqiu gray goats, aligning with the goat genome to obtain 225,417 full-length sequences with an average length of 2138 bp, 207,181 of which were high-quality sequences, accounting for 91.91% of the total [10]. It can be observed that our study obtained more non-chimeric full-length sequences, but the number of high-quality sequences was significantly fewer than that reported by Pan et al., possibly due to factors such as the relatively limited variety of skin cells, analysis software, and parameter settings. The average sequence length was similar, ranging from 2100 to 2400 bp, indicating the conservation of transcript lengths across different goat breeds and tissues.

After further filtering, this experiment obtained 34,253 isoform sequences from high-quality sequences, identifying a total of 10,907 annotated genes and 2834 novel genes through comparison. The identification of these novel genes is particularly important, as they may be associated with unique adaptive traits in Ganxi goats, which can be leveraged to improve breeding programs. Moreover, the discovery of new genes enhances our understanding of the genome’s complexity and opens up new avenues for biomedical research, including the development of therapeutic strategies for wound healing and skin regeneration in both animals and humans. Holland et al. described examples of the asymmetric evolution of duplicated homeobox genes in insects, mollusks, and mammals, suggesting that new genes arise from known genes through asymmetric duplication and evolutionary development [11]. Brunskill et al. conducted a transcriptome analysis of metanephric mesenchyme progenitor cells and their first epithelial derivative nephrons using RNA-Seq, and also discovered a large number of new genes [12]. In summary, the discovery of new genes is of great significance for enhancing our understanding of genomes, revealing biological complexity, and advancing medical research.

In this study, isoform sequences were compared with the above-mentioned public databases, resulting in annotations for 30,460 isoforms in NR, 29,188 in GO, 22,265 in KEGG, 26,842 in Pfam, 30,190 in eggNOG, and 29,729 in Swiss-Prot. GO enrichment analysis revealed that, after redundancy reduction, isoforms were mainly concentrated in cellular processes and biological regulation, cellular components such as cellular anatomy entities and protein-containing complexes, and molecular functions such as adhesion and catalytic activity. KEGG analysis showed that, after redundancy reduction, isoforms were mainly enriched in signal transduction, the immune system, and transport and degradation metabolism signal pathways. The GO and KEGG enrichment pathways were consistent with research areas related to skin function [13,14], reflecting the high correlation and accuracy of enrichment analysis. Additionally, 30,190 isoforms were annotated in eggNOG, primarily in 24 categories, with the highest annotation in unknown functions, followed by signal transduction mechanisms, transcription, post-translational modification, protein turnover, and chaperones. This reflects the incomplete annotation of the goat genome, necessitating further annotation. These data further enrich the annotation information of the goat genome, contributing to a deeper understanding of goat genetic diversity and biological characteristics. This has potential applications in goat breeding, health management, and disease research.

LncRNAs are a class of RNA molecules that do not encode proteins. LncRNAs are associated with the occurrence and development of various diseases, including cancer [15], cardiovascular diseases [16], neurodegenerative diseases [17], etc., and can serve as potential biomarkers or therapeutic targets. Li X et al. used the Ribo Zero RNA sequencing (RNA-Seq) method to analyze the expression profile of LncRNAs in Zhongwei goat skin tissues at 45 days (wavy) and 108 days (straight) after birth, identifying 13,549 LncRNAs, many of which were differentially expressed genes, primarily enriched in the PI3K-Akt signaling pathway and cAMP signaling pathway, which predicted that these LncRNAs may affect hair follicle development and wool fineness by regulating target genes [18]. Zheng Y et al. identified a new lncRNA-000133 from the secondary hair follicles of Cashmere goats and characterized it through ceRNA network analysis. Its potential impact on the induction characteristics of dermal papilla cells was evaluated through overexpression analysis [19]. This study detected 32 known LncRNAs and 256 unknown LncRNAs, further enriching the gene pool for lncRNA research and providing more potential candidates for the function and biological roles of LncRNAs. These new unknown LncRNAs may play important roles in cell signaling, gene expression regulation, chromatin structure, and many other biological processes related to wound healing, highlighting their significance in understanding gene expression regulation and cellular functions.

Alternative splicing events are a significant source of biological diversity and complexity. Through different splicing mechanisms, a gene can encode multiple proteins, which is crucial for the normal functioning of cells and organisms. Kim YE et al. demonstrated that the TJP1-α isoform enhances the assembly of actomyosin stress fibers, thereby promoting cell migration in wound healing experiments [20]. Wang J et al. found that the appearance of the abnormal splicing variant BIN1+12A can counteract the invasion inhibition activity and pro-apoptotic characteristics of BIN1 in non-small-cell lung cancer (NSCLC) [21]. It is evident that alternative splicing events play an important regulatory role in the wound healing process. This experiment identified a total of 12,283 alternative splicing events, expanding the foundational data for genome splicing variant annotation and research on the regulation of splicing variants in wound healing.

In most eukaryotic genes, the transcription process produces a mature mRNA molecule through a specific splicing mechanism, which is then translated into a protein. However, the precursor mRNA of certain genes can undergo different splicing mechanisms, selecting different splicing sites to produce different mRNA splice isoforms, a process known as alternative splicing [22]. Since alternative splicing does not involve permanent changes in genetic information, it is a relatively flexible mechanism in eukaryotic gene expression regulation. Alternative splicing plays a crucial role in regulating gene expression and generating diverse proteomes, making it a key driver of genetic and protein diversity in eukaryotes [23].

Third-generation sequencing technologies greatly facilitate the discovery of more splicing sites, enabling researchers to obtain more gene-specific transcripts and new alternative splicing events [6]. Second-generation sequencing technologies allow for counting and analyzing the expression levels of splicing sites, making more in-depth studies possible due to their cost-effectiveness and efficiency. In this experiment, high-throughput sequencing technology was used to successfully identify hub genes (*CDC20*, *MMP2*, *TIMP1*, and *EDN1*) that play crucial roles in wound healing and cuproptosis processes. Subsequently, we further analyzed the splice variants and expression changes of these genes. This study aims to deepen our understanding of the functions and regulatory mechanisms of these genes in the skin wound healing process. To achieve this, we conducted bioinformatics analyses, focusing particularly on the splice variant events of these genes in order to reveal the regulatory mechanisms between wound healing and cuproptosis.

CDC20 (Cell Division Cycle 20 Homolog) is a protein kinase that plays a crucial role in cell cycle regulation, particularly in the mitotic phase of cell division [24]. Aberrant expression or mutations of CDC20 are associated with cell cycle dysregulation and the development of cancer [25]. Therefore, CDC20 is an important research target in cell biology and cancer research, essential for deepening our understanding of cell division, the pathogenesis of cancer, and potential therapeutic avenues. In this study, three different splice variants of the *CDC20* gene were observed, including variants with changes at the 5′ and 3′ ends, as well as novel variants with new splice sites. Although there were no significant differences in the expression levels of each splice variant within the group, all *CDC20* splice variants showed a significantly increased expression in the skin samples collected 5 days after injury compared to those collected immediately after (0 days), particularly the novel splice variants. Currently, there are no reports related to splice variants of the *CDC20* gene, possibly because its coding sequence (CDS) remains unchanged, thereby not producing new proteins and being overlooked. However, the abnormal expression and mutations of *CDC20* are closely related to human diseases, extensively studied in the field of cancer, where it has been identified as a new therapeutic target for cancer treatment [26,27]. It has also garnered attention in diseases affecting the nervous system [28] and immune system [29]. This study on the identification and splice variants of *CDC20* provides new insights and directions for research into these diseases. Specifically, further exploration of *CDC20* splice variants under different disease conditions can deepen our understanding of their distinct functions and regulatory mechanisms. This deeper understanding of *CDC20*’s role in gene regulation, protein interactions, and cell signaling pathways will enhance our understanding of the pathogeneses of various human diseases such as cancer, neurological disorders, and immune system diseases. Moreover, it offers potential opportunities for developing new therapeutic strategies and biomarkers.

MMP2 (Matrix Metallopeptidase 2), a member of the matrix metalloproteinase family (MMPs), plays a crucial role in degrading collagen and other extracellular matrix proteins to maintain tissue structure and function [30]. MMP2 is involved in various physiological and pathological processes such as tissue remodeling [30], cancer metastasis [31], cardiovascular diseases [32], and arthritis [33]. *MMP2* plays a critical role in tissue remodeling and repair, and it has been identified as a hub gene in various processes such as skin infection [34], skeletal development [35], and cardiac remodeling [36]. These findings align with the results of this study based on a skin wound healing model. This reflects the rationality of experimental design and the high reliability of enrichment analysis and hub gene selection. The activity of MMP2 is tightly regulated, including transcriptional regulation, protein hydrolysis, and inhibition by tissue inhibitors [37]. Aberrant MMP2 activity can lead to tissue destruction and disease progression [38]. Therefore, MMP2 is extensively studied as a significant biomarker and potential therapeutic target [39]. The overactivation of MMP2 is associated with cancer cell invasion and metastasis by degrading the extracellular matrix, facilitating cancer cell penetration through tissue boundaries and entry into the vascular system [31]. Additionally, *MMP2* splice variants may also play different roles in various physiological and pathological processes. In this study, seven different splice variants of the *MMP2* gene were identified, including variants with changes at the 5′ and 3′ ends, novel variants with new splice sites, and a variant with incomplete 3′ splicing. Although there were no significant differences in the expression levels of each splice variant within the group, five *MMP2* splice variants showed significantly increased expression in the skin samples collected 5 days after injury compared to those collected immediately after (0 days), including a variant with a new splice site. *MMP2* splice variants have been widely studied and reported, including splice deletions, changes in splice sites, and N-terminal variants, each potentially playing different roles in physiological and pathological processes. Studies have shown that a splice variant of MMP-2 lacking a signal sequence in human myocardial cells specifically enhances the cleavage of cardiac troponin I during hypoxia-reoxygenation injury [40]. Additionally, *MMP2* splice variants identified in multicenter osteolytic patients may introduce a premature termination codon disrupting the collagen-binding region of *MMP2* [41]. These findings underscore the significant role of *MMP2* splice variants in disease onset and progression, highlighting the importance of further research for understanding disease mechanisms and developing treatment strategies. This study identified two new *MMP2* splice variant forms, providing an additional theoretical basis for splice variant research. The association of *MMP2* with wound healing and cuproptosis, as a core gene identified in screening, suggests that its regulation of splice variant expression holds valuable research potential.

TIMP1 (Tissue inhibitor of metalloproteinases 1) is a metalloproteinase inhibitor that plays a crucial role in inhibiting various metalloproteinases, particularly matrix metalloproteinases (MMPs), thereby preventing the degradation of collagen and other extracellular matrix proteins. It is involved in extracellular matrix degradation and remodeling, contributing significantly to tissue repair and remodeling processes [42]. TIMP1 is implicated in a variety of physiological and pathological processes, including tissue development, wound healing, inflammation, cancer, and cardiovascular diseases [43]. Aberrant expression of TIMP1 has been associated with the development and metastasis of certain cancers [44]. Research on TIMP1 has primarily focused on its gene expression and protein translation levels, with no reported findings on TIMP1 splice variants. In this experiment, splice variants of *TIMP1* were identified and analyzed for their expression levels and gene structures, providing insights into their roles in various physiological and pathological processes. Two new splice variants of *TIMP1* were identified, including one variant with at least one new splice site and one variant with intron retention. Compared to the skin samples collected immediately after injury (day 0), the expression levels of all *TIMP1* splice variants were significantly increased in the samples collected at day 5 post-injury. Among these variants, the splice variant PB.12178.1 was the predominant form and showed a significantly higher expression compared to the splice variant PB.12178.2 in the day 0 samples. This indicates that *TIMP1* splice variants are involved in the regulation of wound healing and possibly apoptosis, mechanisms that warrant further investigation. Research on *TIMP1* not only enhances our understanding of extracellular matrix regulation and metalloproteinase control, but also provides potential targets for the treatment of related diseases.

Endothelin 1 (EDN1, also known as ET-1) is a small peptide produced by endothelial cells that plays crucial biological roles in various physiological and pathological processes such as vascular tension regulation, cell proliferation, and inflammation [45,46]. EDN1 is involved in promoting angiogenesis, which is essential for wound healing and tissue repair processes [47]. It regulates leukocyte migration, the release of inflammatory mediators, and controls the extent of inflammatory responses, thereby facilitating wound healing [48]. An abnormal expression of EDN1 is associated with several diseases, including cardiovascular diseases [49], pulmonary hypertension [50], kidney diseases [51], and certain cancers [52]. Consequently, EDN1 has emerged as a potential target for drug development and therapeutic interventions. Currently, there are no reports on alternative splicing variants of EDN1. In this experiment, three different splice variants of *EDN1* were identified, including variants with changes at the 5′ and 3′ ends, as well as a novel variant with a new splice site. Compared to the skin samples taken immediately after injury (day 0), the qPCR results for *EDN1* on day 5 post-injury showed a significant decrease, with the expression levels of each *EDN1* splice variant also declining in the sequencing results. It can be inferred that *EDN1* splice variants may be involved in the regulation of wound healing and apoptosis induced by copper, but these mechanisms require further investigation. Liu et al., through bioinformatics analysis, explored the key genes and pathways in the formation of corneal scars, identifying *EDN1* as one of seven hub genes involved in the regulation of corneal scar formation [53], which is consistent with the results of this study. This experiment further enriches the understanding of *EDN1* splice variants and their association with copper-induced apoptosis, providing potential targets for the treatment of related diseases.

## 5. Conclusions

In this study, PacBio third-generation sequencing technology was applied to conduct full-length transcriptome sequencing of Ganxi goats’ skin, thereby supplementing and improving the annotation information of the Ganxi goat genome. A total of 926,667 full-length non-chimeric sequences were obtained, optimizing the annotation information of 3794 genomic loci and identifying 2834 new genes, 256 new LncRNAs, 12,283 alternative splicing events, 549 genes with polyadenylation signals, and 112 fusion genes. By integrating transcriptomic data from second- and third-generation sequencing, the splice variant events of hub genes (*CDC20*, *MMP2*, *TIMP1*, and *EDN1*) and changes in the expression levels of these variants in skin samples at day 0 and day 5 post-injury were analyzed. The findings suggest that these splice variants play a regulatory role in wound healing.

## Figures and Tables

**Figure 1 animals-14-03085-f001:**
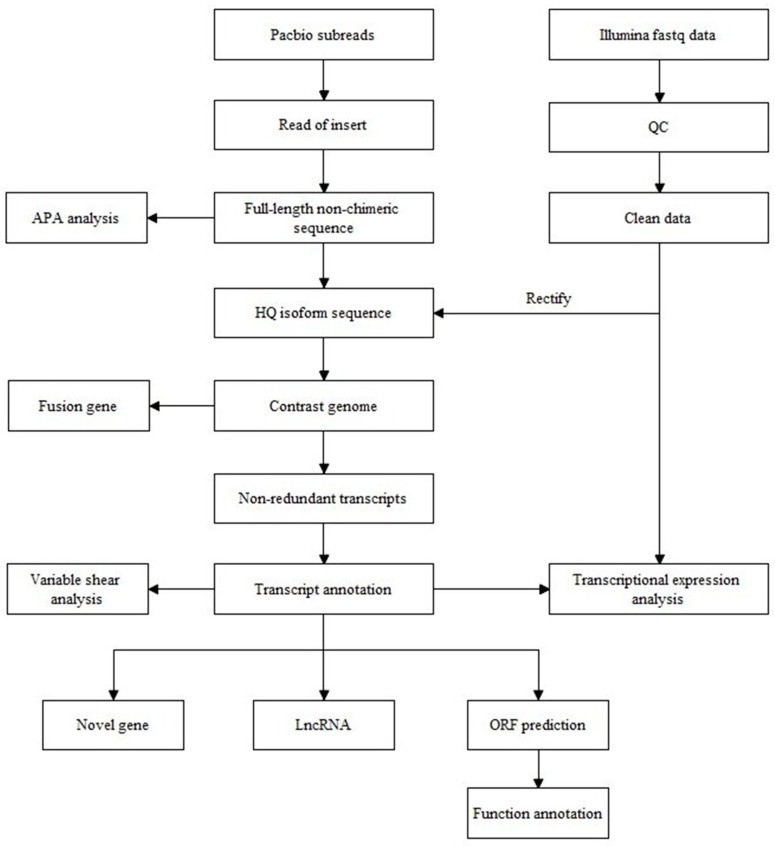
Flowchart of combined second- and third-generation transcriptomics analysis. Note: The information within the boxes represents the items being tested, while the arrows indicate the order of testing.

**Figure 2 animals-14-03085-f002:**
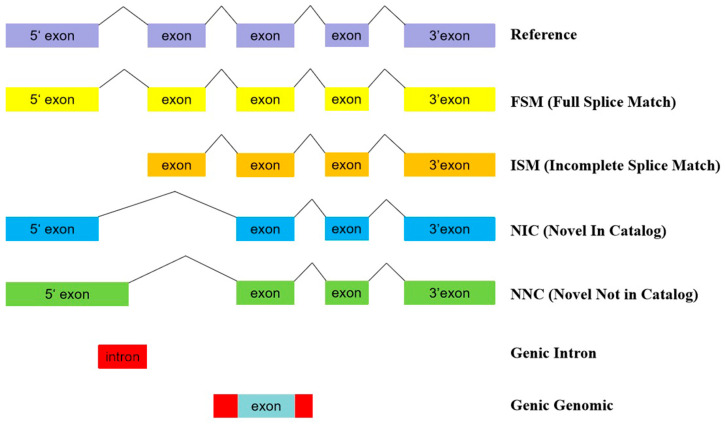
Relationship between different isoforms and reference transcripts.

**Figure 3 animals-14-03085-f003:**
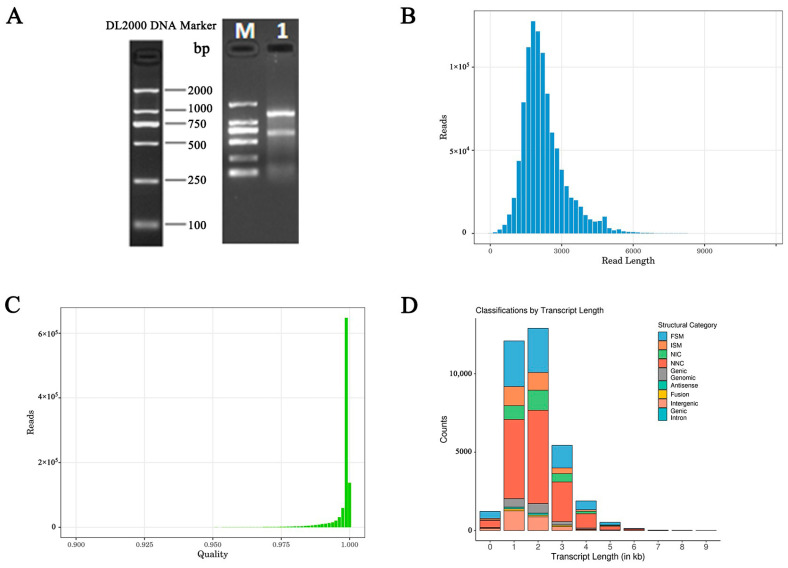
Total RNA quality assessment and sequencing results analysis. (**A**) Agarose gel electrophoresis of total RNA from Ganxi goat skin mixed sample (note: M: DNA Marker 2000; 1: Total RNA after mixing); (**B**) ROI sequence length distribution (note: the horizontal coordinate is the length of ROI, and the vertical coordinate is the amount of ROI); (**C**) quality distribution of ROI sequence (note: the horizontal coordinate is the ROI quality value, and the vertical coordinate is the ROI quantity); and (**D**) distribution of notes on transcripts of different lengths.

**Figure 4 animals-14-03085-f004:**
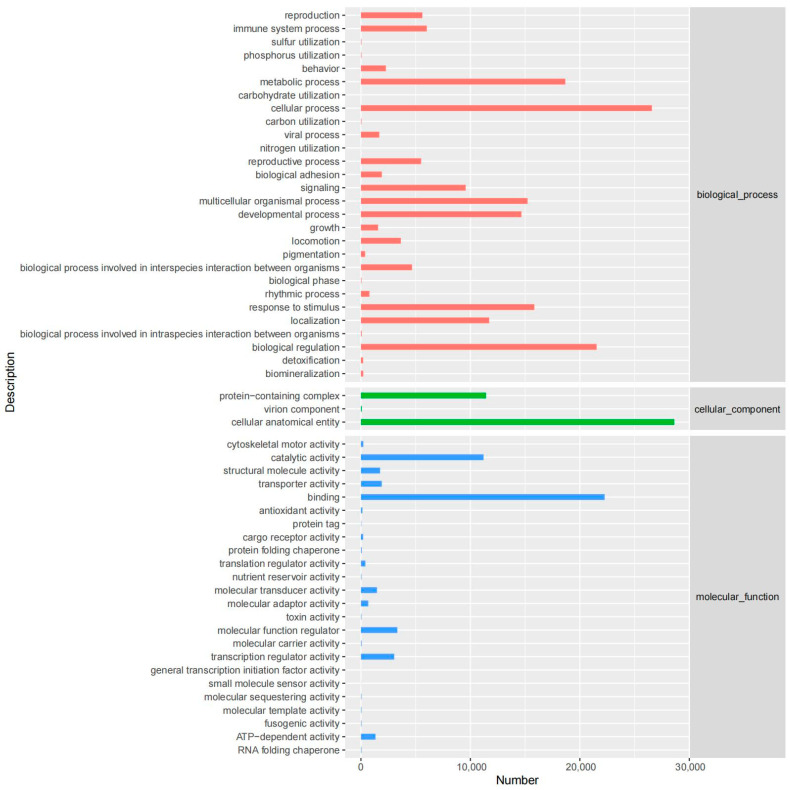
GO annotation statistics.

**Figure 5 animals-14-03085-f005:**
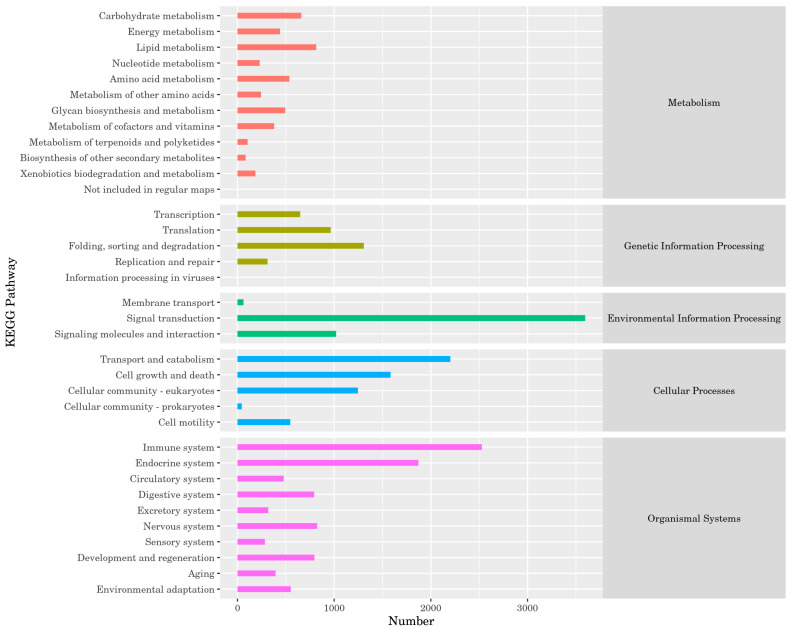
KEGG annotation statistics.

**Figure 6 animals-14-03085-f006:**
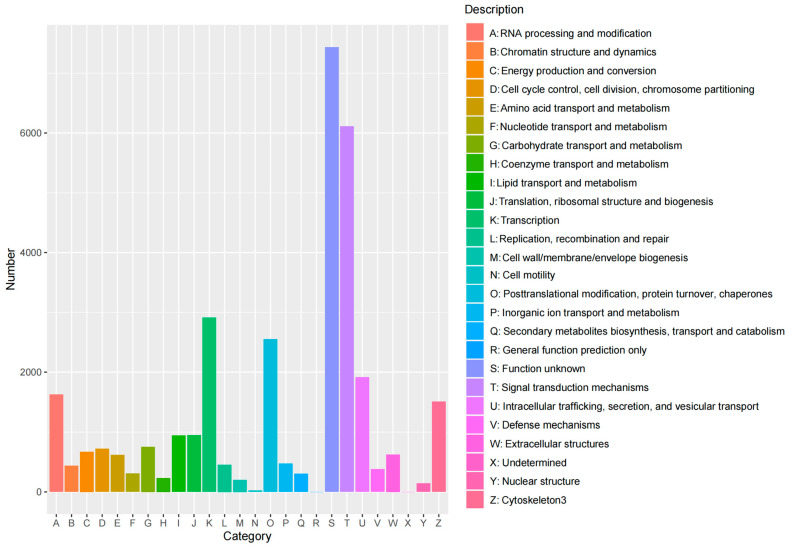
eggNOG annotation statistics.

**Figure 7 animals-14-03085-f007:**
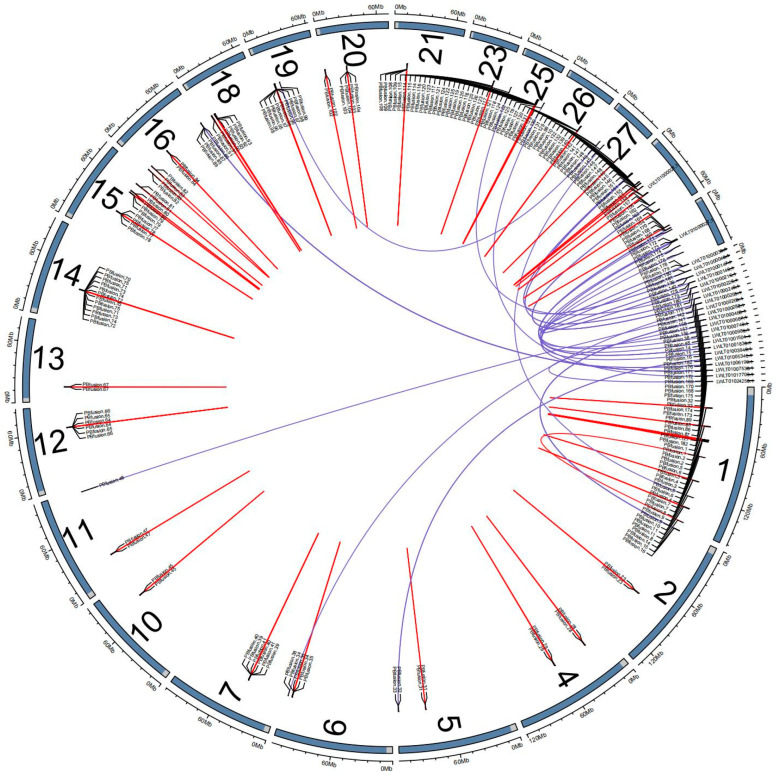
Chromosome distribution of fusion genes. Note: The outermost circle represents the chromosomes; each line within the circle represents a fusion event. The ends of the lines indicate the breakpoint positions of the fusion events. Red lines indicate that the upstream and downstream genes of the fusion gene originate from the same chromosome, while blue lines indicate that they originate from different chromosomes.

**Figure 8 animals-14-03085-f008:**
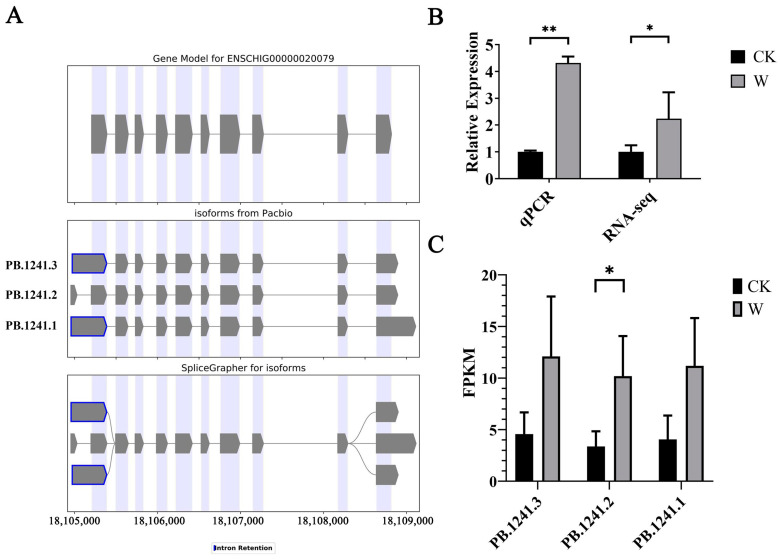
Identification and analysis of *CDC20* alternative splicing events. (**A**) Types of *CDC20* alternative splicing events; (**B**) qPCR validation of *CDC20* transcriptome data; and (**C**) expression analysis of *CDC20* splice variants. Note: * indicating significant differences (*p* < 0.05) and ** indicating extremely significant differences (*p* < 0.01).

**Figure 9 animals-14-03085-f009:**
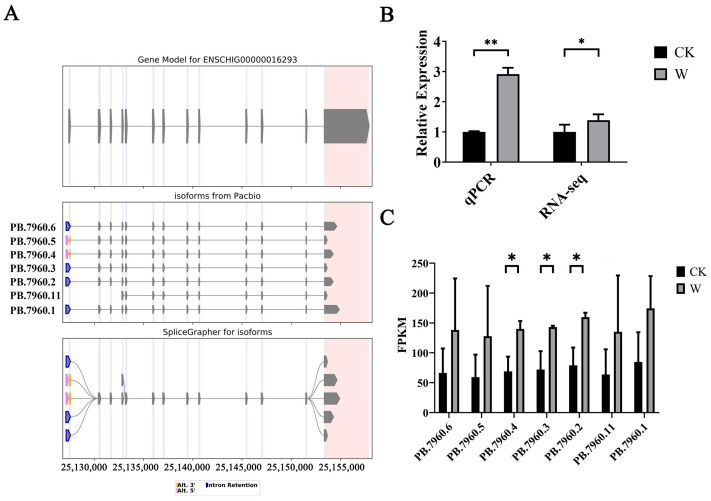
Identification and analysis of *MMP2* alternative splicing events. (**A**) Types of *MMP2* alternative splicing events; (**B**) qPCR validation of *MMP2* transcriptome data; and (**C**) expression analysis of *MMP2* splice variants. Note: * indicating significant differences (*p* < 0.05) and ** indicating extremely significant differences (*p* < 0.01).

**Figure 10 animals-14-03085-f010:**
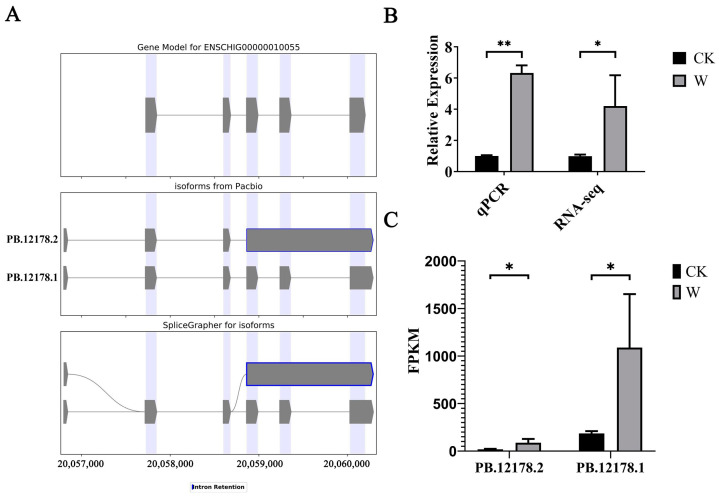
Identification and analysis of *TIMP1* alternative splicing events. (**A**) Types of *TIMP1* alternative splicing events; (**B**) qPCR validation of *TIMP1* transcriptome data; and (**C**) expression analysis of *TIMP1* splice variants. Note: * indicating significant differences (*p* < 0.05) and ** indicating extremely significant differences (*p* < 0.01).

**Figure 11 animals-14-03085-f011:**
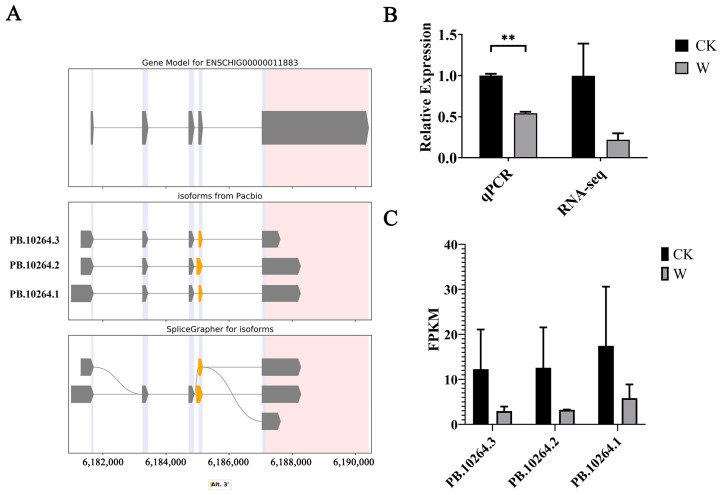
Identification and analysis of *EDN1* alternative splicing events. (**A**) Types of *EDN1* alternative splicing events, Yellow represents exons with -3 ′alterations; (**B**) qPCR validation of *EDN1* transcriptome data; and (**C**) expression analysis of *EDN1* splice variants. Note: ** indicating extremely significant differences (*p* < 0.01).

**Table 1 animals-14-03085-t001:** Primer information.

Primer Name	Sequence (5′ to 3′)	Number of Bases	Product (bp)
GAPDH-F	ATGGTGAAGGTCGGAGTGA	19	152
GAPDH-R	TGGGTGGAATCATACTGGAA	20	
CDC20-F	AGACCTTCACCCAGCATCA	19	146
CDC20-R	ATCCACGGCACTCAGACAG	19	
MMP2-F	TGGTTTCCTCTGGTGTTCC	19	54
MMP2-R	CCATACTTGCCGTCCTTGT	19	
TIMP1-F	CAGAATCGCAGTGAGGAGTTT	21	145
TIMP1-R	CACAACCAGCAGCATAGGTC	20	
EDN1-F	TCATCTGGGTCAACACTCC	19	126
EDN1-R	TACACTGGCATCTCTTCCTG	20	

**Table 2 animals-14-03085-t002:** qRT-PCR reaction system.

Reagent	System
2 × SYBR real-time PCR pre mixture	10 μL
10 μM PCR-specific primer F	0.4 μL
10 μM PCR-specific primer R	0.4 μL
cDNA	1 μL
RNase free ddH_2_O	Up to 20 μL

**Table 3 animals-14-03085-t003:** Detection results of total RNA Agilent 2100 Bioanalyzer in skin of Ganxi goats.

Sample	Sample Name	Concentration (ng/μL)	Total Amount (μg)	OD260/280	OD260/230	RIN Value
Skin	Mixed Total RNA	474.47	17.081	2.051	2.051	9

**Table 4 animals-14-03085-t004:** Statistics of Pacbio sequencing data.

Sample	Number of ROI Sequences	Total ROI Bases	Average ROI Sequence Length	Average Quality Value of ROI Sequences	Average Passes of All Sequences
Skin	1,010,225	2,367,860,756	2343	1	29.38

**Table 5 animals-14-03085-t005:** Structure information of *CDC20* splicing variants.

Alternative Splicing Variant	Total Length (bp)	Number of Exons	CDS Length (bp)	Aligned Genome	Splicing Form
PB.1241.1	2046	10	1500	Fully spliced match	Variable 5′ and 3′ ends
PB.1241.3	1817				
PB.1241.2	1658	11		Unmatched	At least one novel splice site

**Table 6 animals-14-03085-t006:** Structure information of *MMP2* splicing variants.

Alternative Splicing Variant	Total Length (bp)	Number of Exons	CDS Length (bp)	Aligned Genome	Splicing Form
PB.7960.1	3731	13	1986	Fully spliced match	Variable 5′ and 3′ ends
PB.7960.2	3110				
PB.7960.3	2457				
PB.7960.6	3435				
PB.7960.4	2923	14	1755	Unmatched	At least one novel splice site
PB.7960.5	2294				
PB.7960.11	1650	10	1140	Partial splice match	−3′ end

**Table 7 animals-14-03085-t007:** Structure information of splicing variants of *TIMP1*.

Alternative Splicing Variant	Total Length (bp)	Number of Exons	CDS Length (bp)	Aligned Genome	Splicing Form
PB.12178.1	766	6	624	Unmatched	At least one novel splice site
PB.12178.2	1677	4	438		Intron retention

**Table 8 animals-14-03085-t008:** Structure information of *EDN1* splicing variants.

Alternative Splicing Variant	Total Length (bp)	Number of Exons	CDS Length (bp)	Aligned Genome	Splicing Form
PB.10264.1	2342	5	609	Fully spliced match	Variable 5′ and 3′ ends
PB.10264.3	1393				
PB.10264.2	2107		411	Unmatched	At least one novel splice site

## Data Availability

We have uploaded the sequencing raw data for this study to the SRA database (https://www.ncbi.nlm.nih.gov/sra/PRJNA922517, URL (accessed on 20 06 2024)).

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
