# Peer review of "Combined Analysis of Second- and Third-Generation Transcriptome Sequencing for Gene Characteristics and Identification of Key Splicing Variants in Wound Healing of Ganxi Goat Skin"

_animals, 2024, doi:10.3390/ani14213085_

Round 1
Reviewer 1 Report (Previous Reviewer 2)
Comments and Suggestions for Authors
The revised manuscript is more informative for the readers, it could be published in its present form.
Author Response
Comments 1: The revised manuscript is more informative for the readers, it could be published in its present form.
Response 1: Thank you for your positive feedback on our revised manuscript. We are glad to hear that the revisions have made the manuscript more informative for readers. We appreciate your time and thoughtful review.

Reviewer 2 Report (Previous Reviewer 4)
Comments and Suggestions for Authors
I appreciate the authors' changes and corrections in the revised articles. However, I would like to add a few points to improve the article before publication.
Comments
1. The authors have given the answer about the importance of splice variants and previous knowledge on wound healing. I suggest that the authors should include this information in the Introduction, which was presented as a response to the reviewer.
2. I suggest that the authors double check the mention of the EDN1 response in the text.
3. I cannot find the supplementary materials with this revised version of the article.
4. The supplementary materials/files should be mentioned in the article text.
Author Response
Please see the attachment

This manuscript is a resubmission of an earlier submission. The following is a list of the peer review reports and author responses from that submission.
Round 1
Reviewer 1 Report
Comments and Suggestions for Authors
Through the combined analysis of second- and third-generation transcriptome sequencing, the authors identified splicing variants associated with wound healing as well as the enhancement of genome annotation in the skin of Ganxi goats. However, since samples were collected on days 0 and 5 post-injury, the analysis could have included differential expression analysis, using newly discovered annotations. Additionally, it is suggested that the core content of the manuscript be simplified for clarity and conciseness. The inclusion of too many tables and figures may be counterproductive, and the interpretation of the observed results is currently insufficient. Therefore, I do not think this manuscript is suitable to be published in Animals.
1. (lines 66-67) Previous study showed cuproptosis-related genes, so it is necessary to explain what kind of relationships it has with ferroptosis.
2. Library construction should be written before sequencing in the method part.
3. QC method of data for Illumina sequencing is also required.
4. The method of identifying Novel lncRNAs is too simplified.
5. I think qPCR validation method should be located before data statistical analysis.
6. I think that “Ch” indicates an abbreviation for Capra hircus in most tables but the word looks not suitable. Please revise appropriately.
7. Too many tables and figures are shown. Some less significant tables and figures need to be changed to supplementary.
8. Why did the authors focus only on the splicing variants of the 4 genes? Since the whole-length sequencing was performed rather than target sequencing, the author could also have found new candidates through differential expression analysis.
9. The authors should provide location information of newly identified features (novel genes, lncRNAs, etc.) based on the specific reference genome.
Reviewer 2 Report
Comments and Suggestions for Authors
The aim of the manuscript is to identify splice vaiants of core would healing genes in a specific breed of goat the Gangxi. The applied method is adequate and up-to-date. The analysis of the data obtained with Pac-Bio third generation sequencing of the skin transcriptome is correct and clearly presented. The database presented could be a starting point to answer biological questions, but in its present form it does not go beyond presenting the results of the above mentions sequencing in two different time points. The most interesting part is the identification and expression analysis of the four HUB genes in wound healing, which again could be a starting point for deeper understanding their exact role in theis process. The manuscript has several phrases repeated word by word in different sections, which sould be carefully examined in deleted where it is appropriate. Examples: line 650-658,677-684, 700-707 etc. Lines 609- 620 are advertisement for PacBio sequencing should be removed. 3.3.6 section is again repetitive from line 457-466! This text was already present in 2.5.7. The text on page 8. lines 250-260. was probably copied from somewhere, because the manusript does not have Chapter Three!!
Reviewer 3 Report
Comments and Suggestions for Authors
Manuscript technically sounds good. However, throughout the manuscript mainly methods were mentioned and results focused on the identified genes. Manuscript should be revised in the context of why wound healing or skin regeneration is important and how this could be used in livestock breeding or medical purposes. Therefore, introduction and discussion should be revised dramatically in order to give information about why this biology was selected and how findings could be used further research.
Reviewer 4 Report
Comments and Suggestions for Authors
The study, authored by Yang et al., investigated the transcriptome sequencing and identification of crucial splicing variants of hub genes related to wound healing process of Ganxi goat skin. The study presented clearly by the authors and the manuscript was written well. Since, there are few concerns need to clarify and correct for the improvement of the article for the understanding of the present study by the readers.
Major comments
1. Introduction Line 66-70: The given splice variant genes of hub genes were reported in any other ruminant species? What are the importance of the genes and why it is significant in the ganxi goat species?
2. The methodology part was presented well, but the figure 1 title showed not enough information on understanding the flowchart. what type of analysis? mention like sequence analysis and annotaion of genes. Mention, what the flowchart depicting ? is it step by step process ? what is the classification of another parallel process? Which function annotation? Gene ontology?
3. The authors used several softwares for statistics and sequence annotation process. Line 149: SQANTI2 and Line 205: SpliceGrapher. Missing information on version and whether these softrware are free or commercial ?
4. The authors have provided several tables in the result part such as table 5, 6, 7, 8, 9, 10, 11, 12, 13 are repeted information as presented in the assosiated main text. These tables not essential and not giving a new information other than the text and which are already well explained in the text on number counts. I suggest the authors to keep them in the supplimentary file.
5. Is there any GO function related to the wound healing activities?
6. Figure 8 was not well explained in the legend. What are the red dots representing? map quality is not well explained and the scores, enlarge the number fonts, which is present in the external.
7. The same information of table 14 given in the before paragraph.
8. Line 517: the new splice variant PB:1241.2 or PB:1241.3? Because, there is a slight increase in W of PB:1241.3 in figure 9C.
9. Line 549: why the PB.7960.6 was not included, which showed similar increase as other selected splice varients in figure 10C.
10. Why the authors included EDN1 splicing variants? which showed decrease expression on all the thre variants. How it can be participate in the wound healing process?
11. Line 834: The EDN1 splice variant decreased in the sample after 5 post injury and no significant difference observed. Line 836: How the authors ensured this gene may participate in the regulation of wound healing and apoptosis process?
Minor comments
1. The authors needs to change all the genus and species names in "italic" fonts, specifically in simple summary and line 386-391. Please check throughout the manuscript.
2. Abstract line 32-34: rewrite phrase: Three splice variants forms was identified in the both CDC20 and EDN1 genes.
3. Line 125: Expansion for CCS reads?
4. The authors must follow the denomination (,) on numbers, line 395-402 given but not followed in the tables.